# Automatic Tracking of Weak Acoustic Targets within Jamming Environment by Using Image Processing Methods

**Fan Yin** [1,2], **Chao Li** [1,2,*], **Haibin Wang** [1,2] **and Fan Yang** [3]

1 State Key Laboratory of Acoustics, Institute of Acoustics, Beijing 100190, China; yinfan0120@foxmail.com (F.Y.); whb@mail.ioa.ac.cn (H.W.)
2 School of Electronic, Electrical and Communication Engineering, University of Chinese Academy of Sciences, Beijing 100049, China
3 Laboratory of ImViA, University of Burgundy-France-Comté, 21078 Dijon, France; fanyang@u-bourgogne.fr
* Correspondence: chao.li@mail.ioa.ac.cn

**Abstract:** Bear time records, which are the accumulations of spatial spectrum estimates on the time axis, are often employed for passive sonar information processing. Multi-target jamming is a common difficulty in this approach due to the constraints of Rayleigh limit, and neither the conventional beamforming (CBF) nor minimum variance distortionless response (MVDR) technique can handle it well. This work presents a post-processing tracking framework based on visual pattern recognition algorithms to track weak acoustic targets within jamming environments, which includes target motion analysis, matched filtering, and principal component analysis-based denoising, and we call this 'P-Gabor' algorithm. The simulations and sea-trial experiments show that the proposed method can track a weak target successfully under −23 dB (signal-to-interference ratio) SIR, which is more effective than the references, especially in terms of using real-world data from sea trials. We further demonstrate that the method also has stable tracking performance at even −25 dB SNR (signal-to-noise ratio) circumstances.

**Keywords:** matched filtering; jamming targets; target tracking; Gabor filtering; direction of arrival; principal component analysis; P-Gabor

## 1. Introduction

In underwater target detection and tracking, acoustic waves are the best known mechanical waves that can operate long distance and at low loss underwater, so they are widely used in the field of ocean exploration. In order to obtain higher detection gain for acoustic waves, we often use array signal processing technology to detect targets through an array of multiple array elements, so as to detect farther and weaker targets.

One of the pretty significant study subjects in array signal processing, commonly known as spatial spectrum estimation, is direction of arrival (DOA) [1]. When numerous DOA spectrograms on the time axis are gathered, a diagram of bearing time records (BTR) is produced. However, BTR tracking is frequently thwarted by jamming targets. The traditional BTR tracking technology is based on the maximum energy tracking method, so when the energy of the target of interest is covered by the interference target, the target of interest will not be tracked successfully. As a result, it is vital to devise a strategy to mitigate the effects of powerful jamming targets.

Due to the Rayleigh limit, it is too hard to track the target with strong interference using a traditional tracking method on the BTR. In the far-field case, the distribution of acoustic wave intensity received by the sonar array without considering its breadth is:

$$I = I_0 \left( \frac{\sin \alpha}{\alpha} \right)^2 \tag{1}$$

where $I_0$ is the normalized maximum intensity amplitude and $\alpha = \frac{\pi a}{\lambda}\theta$, $a$ is the array aperture length, $\lambda$ is the wavelength, and $\theta$ is the resolution angle approximately $\theta \approx \sin\theta$. So we have

$$\theta = \arcsin\frac{\lambda}{a} \qquad (2)$$

According to Equation (2), the sonar array resolution is limited by the target wavelength and array aperture length, and the side lobe energy of strong jamming targets leaks out, thereby masking weak targets. More specifically, as seen in Figure 1a, when the initial null point of one target's wave intensity distribution is lower than the global optimal point of another target, it is only resolvable between those two targets. Otherwise, the sonar array will have difficulty distinguishing between them, especially if the energy of interested target is smaller than that of the strong interfering target (see Figure 1b).

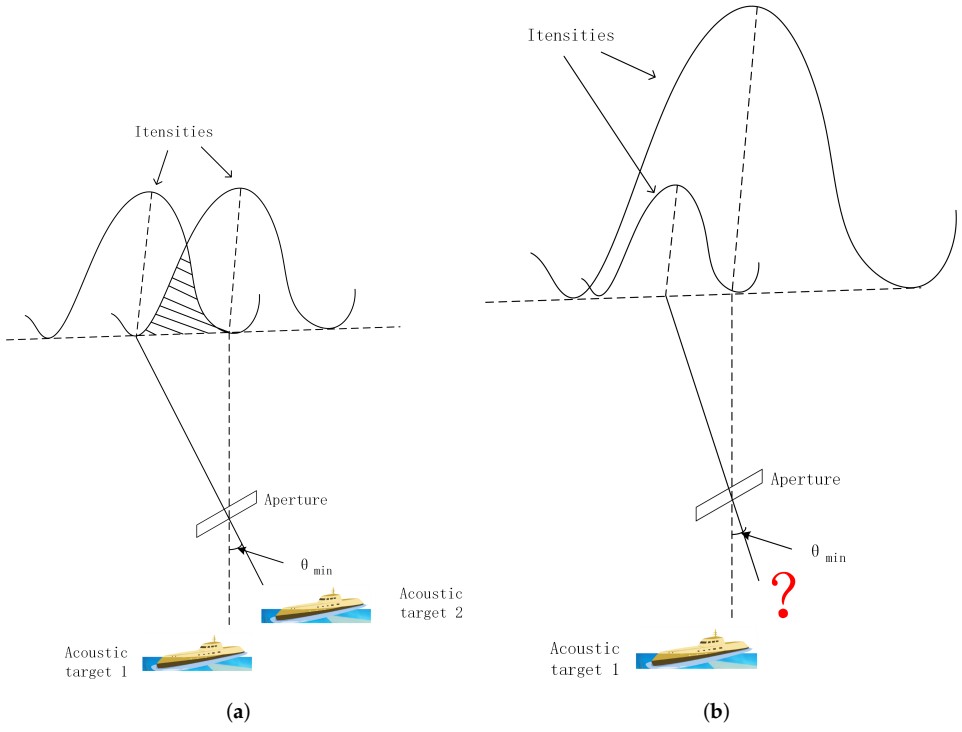

**Figure 1.** Explanation of Rayleigh Limit. (**a**) Distinguishable Target; (**b**) Indistinguishable Target with strong jamming target.

As previously stated, traditional tracking methods are not suitable for environments where there are strong interfering targets in the same direction as the weak target of interest. To mitigate the effects of jamming targets, a new target-tracking technology is urgently needed.

Many studies have recently been conducted to address multi-target jamming issues in sonar array applications. To obtain better tracking performance and stability, F. Yin et al. [2] present a signal–noise separation-based denoise approach that has been applied in passive sonar detecting regions; Escot Bocanegra et al. [3] use principal component analysis in radar systems to obtain high resolution range profiles, but they mainly pay attention to the denoising part rather than jamming target environment. X. Li et al. [4] provide two types of observation modes based on unscented extended Kalman [5] and Kalman filter [6]: (a) static multi observation station and (b) movable single observation station. The experiment findings indicate that the target may be followed stably in the presence of a high interference background. However, it is assumed that the target and the observation station are both at the same depth underwater in two-dimensional space, which is overly utopian and ignores other more typical scenarios. To increase the tracking impact, M. Sheng et al. [7] present

a multi-target data association approach based on a clustering cloud model; it can follow data in real time with great accuracy. As this algorithm is mainly applied in autonomous underwater vehicles (AUVs), the real-time processing ability should be improved in future. G. Wang et al. [8] propose an underwater target-detection and tracking system based on a multi-sensor array in order to increase target-detection capabilities. However, in the process of iterative convergence, it is necessary to consider whether the local optimal solution is the global optimal solution. M. S. Arulampalam et al. [9] provide theoretical guidance for particle-filter tracking in a nonlinear non-Gaussian environment. L. Xu and C. Liu [10] present a particle-filter algorithm based on weight and hypothesis test to determine the start and end of a target trajectory. The experimental findings show that the proposed technique can enhance weak target tracking in a multi-target jamming environment. However, when the number of particles is enormous, the particle-filter method takes a long time. S. Kim et al. [11] present a method based on the graphics processing unit (GPU) to speed the particle filter operation to estimate the changing location of the target, which is well suited to scenes requiring real-time processing. The findings demonstrate that the speed has improved, and the tracking of the target's quick shift trajectory performs well. Z. Wang and Fuhu Chen [12] propose a unified method for target detection and tracking using energy information, which can detect and track the weak targets correctly. Y. Yang and Y. Zhang [13] propose a novel approach for localizing wideband signals in a high-interference environment based on sparse spectrum fitting and a matrix filter with nulling, which is beneficial for passive sonar direction-of-arrival estimation [14,15]. In both simulated and experimental results, the proposed sparse spectrum fitting approach is more resistant to the choice of regulatory parameters and has a lower sidelobe level. However, those two algorithms do consider jamming-target tracking. Yang T. C. [16] employs a kind of signal–noise separation and deconvolution to detect a weak target in the presence of high target interference. The results reveal that it is not able to detect weak targets in low SNR circumstances. Zhang G. P. et al. [17] propose a multi-Bernoulli filter for tracking DOA of targets using sensor array; this allows more accurate multi-target state estimation under jamming circumstances. Li X. et al. [18] propose a bearing-only target tracking method to track multi-targets at the same time. Xiao Chen et al. [19] propose the probability hypothesis density (PHD) and cardinalized probability hypothesis density (CPHD) algorithms based on a novel detection probability to track multi-targets. However, the experiment results of the above three algorithms have no real sea-trial data. Haining W. et al. [20] propose a data-association method based on block association, which realizes correct association for target trajectory crossing, target long-time interruption, etc, but the weak-target-tracking ability is not good.

Our research focuses on how to improve the weak-target-tracking performance under strong jamming-target circumstances by using a post-processing framework. Previously, we improved acoustic-target-bearing trajectories by combining image-processing methods in the post-processing framework of bearing time records [2]. We also presented a solution to suppress the interference of jamming targets on the 2021 China Automation Congress (CAC2021) [21], which is based on linear interpolation template matching, to address the issue of multi-target jamming.

For better tracking performance, this paper improves and supplements our previous work of reference [21], and then unifies Gabor filtering, template matching, and principal component analysis to denoise and suppress jamming targets. The experiment in this work is conducted by using simulated and sea-trial data, and the proposed method is evaluated by comparing with the conventional tracking methods and the reference method presented in [21]. Ultimately, both of the simulation and sea-trial experiments show good performance when tracking weak targets under a jamming-target environment.

The main contributions of this paper include:

(a) At present, the research in the field of BTR post-processing is not sufficient. In this paper, a new BTR post-processing architecture is established, which can be used to improve the resolution and tracking capability of BTR.

(b)     To address the problem of tracking failure caused by BTR image-level noise inter-ference, a real-time PCA processing method is introduced, and a BTR trajectory en-hancement method is proposed to improve the stability of traditional target-tracking methods.

(c)     To address the problem of insufficient resolution caused by the Rayleigh limit of BTR, an image enhancement algorithm to improve the resolution of BTR is proposed by combining the target motion analysis with the Gabor filtering method.

(d)     To acquire a higher matching gain, the template parameters are set using the coherent matching of Gabor's zero point distribution and Rayleigh limit zero point distribution.

(e)     The feasibility of the above method is verified by simulation and sea-trial data.

The remainders of this paper is as follows: Section 2 describes the proposed tracking method in detail; Section 3 presents and analyses the experimental results; finally, some discussion and conclusions are presented in Section 4.

## 2. Automatic Tracking Method of Weak Acoustic Targets within Jamming Environment

This section describes the proposed algorithm—P-Gabor, as seen in Figure 2. Using principal component analysis (PCA), we first denoise the original BTR to improve tracking performance as pre-processing, then we obtain a relatively clean BTR diagram as shown in Figure 2a. The target motion-analysis parameters are then derived using prior linear fitting information as shown in Figure 2b, and those green dots are the discrete track points for each snapshot. Then we use the prior estimated parameter information to generate a template by connecting with the Gabor function as shown in Figure 2c,f. Next, we cut the region of interest (ROI) of BTR as shown in Figure 2d to wait for the matching process; the enlarged jamming part is shown in Figure 2e. Finally, the target bearing is estimated by matching the templates to the target trajectories, as shown in Figure 2f–h. This method is only implemented in the post-processing part, which is the processing unit that previous researchers often neglected in underwater acoustic target detection.

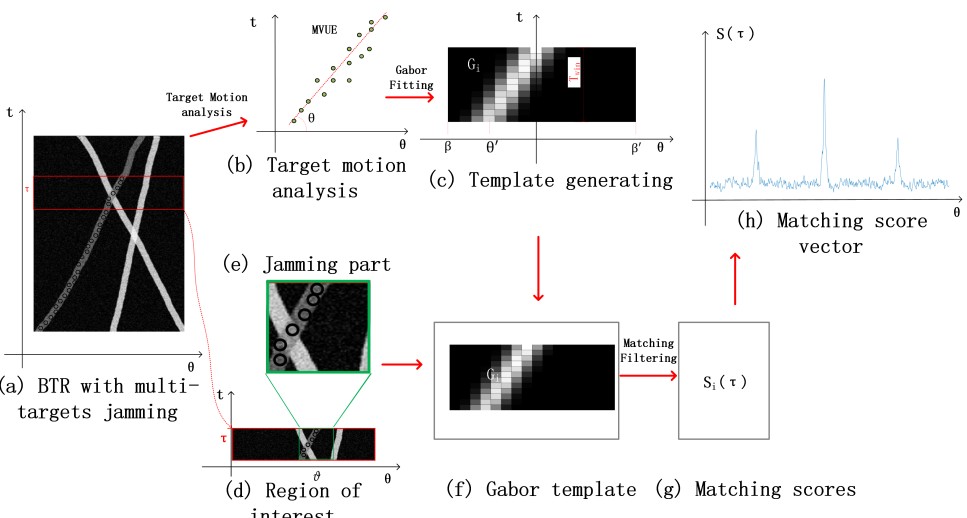

**Figure 2.** Frame of the proposed method after denoising.

### 2.1. De-Nosing

The PCA method is used in this study to denoise BTR diagrams. There exists a trans-forming matrix $P$ in a sample matrix $X = [x_1, x_2, \ldots, x_m]^T$ in $n$ dimensions that can lower its dimension number from $n$ to $l$ by transforming $Y = PX$, where $Y = [y_1, y_2, \ldots, y_m]^T$. PCA enables the reduction of information loss by dimensionality reduction. It can reduce the correlation between samples in order to keep the data diverse in each dimension; this is

accomplished by maximizing variance across all dimensions. The covariance matrix of the reduced dimension sample set $Y$ is

$$C_Y^{l \times l} = \frac{1}{m} YY^T = \begin{pmatrix} \lambda(y_1, y_1) & \cdots & \lambda(y_1, y_l) \\ \vdots & \ddots & \vdots \\ \lambda(y_l, y_1) & \cdots & \lambda(y_l, y_l) \end{pmatrix} \tag{3}$$

where the diagonal elements are the variances of the random variables, and the non-diagonal elements are the covariances between the random variables. According to the principle of PCA, the covariance should be zero, whereas the variance should be non-zero, which means the covariance matrix is a diagonal matrix.

Substitute $Y = PX$ into the covariance matrix $C_Y^{l \times l}$

$$\begin{aligned} C_Y^{l \times l} &= \frac{1}{m} YY^T = \frac{1}{m} PX(PX)^T \\ &= P \frac{1}{m} XX^T P^T = P_{l \times n} C_X^{l \times l} P_{l \times n}^T \end{aligned} \tag{4}$$

where $C_X^{l \times l}$ is the covariance matrix of $X$. Compute the eigenvalue decomposition of $C_X^{l \times l}$

$$\Sigma_{n \times n} = U_{n \times n} C_X^{l \times l} U_{n \times n} \tag{5}$$

Equations (4) and (5) demonstrate that the transformation matrix $P_{l \times n}$ is composed of the first $l$ eigenvectors of matrix $C_X^{l \times l}$ in rows. The principal component analysis therefore can be performed by computing the eigenvalue decomposition of $C_X^{l \times l} = \frac{1}{m} XX^T$.

The singular decomposition (SVD) of rectangle matrix $A \in R^{m \times n}$ is

$$A_{m \times n} = U_{m \times m} \Sigma_{m \times n} V_{n \times n}^T \tag{6}$$

where $U_{m \times m}$ and $V_{n \times n}$ are orthogonal matrices, $\Sigma_{m \times n} = diag(\sigma_1, \ldots, \sigma_n)$ is a diagonal matrix and $\sigma_i$ is the singular value of $A_{m \times n}$. Then,

$$\begin{aligned} A_{m \times n}^T A_{m \times n} &= (U_{m \times m} \Sigma_{m \times n} V_{n \times n}^T)^T U_{m \times m} \Sigma_{m \times n} V_{n \times n}^T \\ &= V_{n \times n}^T) \Sigma_{m \times n}^T U_{m \times m}^T U_{m \times m} \Sigma_{m \times n} V_{n \times n}^T = V_{n \times n} (\Sigma_{m \times n})^2 V_{n \times n}^T \end{aligned} \tag{7}$$

where $\Sigma_{m \times n}^2$ is a diagonal matrix composed of $A_{m \times n}^T A_{m \times n}$'s eigenvalues $\lambda_i$. It can also be seen that the singular value $\sigma_i$ of $A_{m \times n}$ is the square root of $A_{m \times n}^T A_{m \times n}$'s eigenvalue $\lambda_i$. Therefore, either PCA or SVD can be effected by computing the eigenvalue decomposition of a real symmetric matrix. Let $A$ be $\frac{X^T}{\sqrt{m}}$; it yields

$$AA^T = \frac{X^T}{\sqrt{m}}^T \frac{X^T}{\sqrt{m}} = \frac{1}{m} XX^T \tag{8}$$

Due to this equivalence relation, the PCA problem can be solved via SVD, which is more efficient because it avoids the calculation of $AA^T$.

### 2.2. Target Motion Analyzing (TMA)

This paper's technique initially collects some snapshots of trajectory information of interested target, then obtains the TMA parameters like slope and intercept by analyzing the prior trajectory. Normally, the underwater acoustic target varies slowly in a short time; if the bearing of the target at time $t_i$ is $\theta_i$, it can be thought of as linear $\Delta t$ over a specific short period of time. The trajectory of the interested target can be described as $t_i = k\theta_i + b$, where $k$ is the slope and $b$ is the intercept to be estimated. Let $\boldsymbol{\alpha} = (k, b)^T$ represent the parameter vector, so we have $\mathbf{T} = \mathbf{H}\boldsymbol{\alpha} + \mathbf{N}$, where $\mathbf{T} = [t_1, t_2, \cdots, t_n]^T$, $\mathbf{H} = \begin{bmatrix} \theta_1 & \theta_2 & \cdots & \theta_n \\ 1 & 1 & \cdots & 1 \end{bmatrix}^T$,

and **N** is the noise vector with Gaussian distribution. Then calculate its Cramer–Rao Lower Band (CRLB) to obtain the minimum error.

According to the knowledge of probability theory, it is known that

$$\int_{-\infty}^{+\infty} f(t, \boldsymbol{\alpha})dt = \mathbf{I} \tag{9}$$

where $f(t, \boldsymbol{\alpha})$ is the probability density function including two variables $t$ and $\boldsymbol{\alpha}$. Therefore, we have

$$\frac{\partial}{\partial \boldsymbol{\alpha}} \int_{-\infty}^{+\infty} f(t, \boldsymbol{\alpha})dt = \int_{-\infty}^{+\infty} \frac{\partial}{\partial \boldsymbol{\alpha}} f(t, \boldsymbol{\alpha})dt = \mathbf{0} \tag{10}$$

Let $\hat{\boldsymbol{\alpha}}$ be the unbiased estimation of $\boldsymbol{\alpha}$:

$$\int_{-\infty}^{+\infty} \hat{\boldsymbol{\alpha}} f(t, \boldsymbol{\alpha})dt = E(\hat{\boldsymbol{\alpha}}) = \boldsymbol{\alpha} \tag{11}$$

where $E(*)$ returns the expectations of the input variables.

From Equation (11), the following equation can be derived:

$$\frac{\partial}{\partial \boldsymbol{\alpha}} \int_{-\infty}^{+\infty} (\boldsymbol{\alpha} f(t, \boldsymbol{\alpha}))dt = \frac{\partial}{\partial \boldsymbol{\alpha}} (E(\hat{\boldsymbol{\alpha}})) = \frac{\partial}{\partial \boldsymbol{\alpha}} (\boldsymbol{\alpha}) = \mathbf{I} \tag{12}$$

Equation (12) can be also expressed as

$$\begin{aligned}\frac{\partial}{\partial \boldsymbol{\alpha}} \int_{-\infty}^{+\infty} (\boldsymbol{\alpha} f(t, \boldsymbol{\alpha}))dt &= \int_{-\infty}^{+\infty} \frac{\partial}{\partial \boldsymbol{\alpha}} (\boldsymbol{\alpha} f(t, \boldsymbol{\alpha}))dt \\ &= \int_{-\infty}^{+\infty} (\mathbf{I} * f(t, \boldsymbol{\alpha}) + \boldsymbol{\alpha} \frac{\partial}{\partial \boldsymbol{\alpha}} f(t, \boldsymbol{\alpha}))dt = \mathbf{I} + \int_{-\infty}^{+\infty} \boldsymbol{\alpha} \frac{\partial}{\partial \boldsymbol{\alpha}} f(t, \boldsymbol{\alpha})dt\end{aligned} \tag{13}$$

Therefore, we have

$$\int_{-\infty}^{+\infty} \boldsymbol{\alpha} \frac{\partial}{\partial \boldsymbol{\alpha}} f(t, \boldsymbol{\alpha})dt = \mathbf{0} \tag{14}$$

Computing the difference of Equations (11) and (14), we obtain

$$\int_{-\infty}^{+\infty} (\hat{\boldsymbol{\alpha}} - \boldsymbol{\alpha}) \frac{\partial}{\partial \boldsymbol{\alpha}} f(t, \boldsymbol{\alpha})dt = \mathbf{I} \tag{15}$$

Because $\frac{\partial}{\partial \boldsymbol{\alpha}} f(t, \boldsymbol{\alpha}) = \frac{\partial}{\partial \boldsymbol{\alpha}} (\ln f(t, \boldsymbol{\alpha})) f(t, \boldsymbol{\alpha})$, Equation (15) can be rewritten as

$$\begin{aligned}&\int_{-\infty}^{+\infty} (\hat{\boldsymbol{\alpha}} - \boldsymbol{\alpha}) \frac{\partial}{\partial \boldsymbol{\alpha}} (\ln f(t, \boldsymbol{\alpha})) f(t, \boldsymbol{\alpha})dt \\ &= \int_{-\infty}^{+\infty} (\hat{\boldsymbol{\alpha}} - \boldsymbol{\alpha}) \sqrt{f(t, \boldsymbol{\alpha})} \frac{\partial}{\partial \boldsymbol{\alpha}} (\ln f(t, \boldsymbol{\alpha})) \sqrt{f(t, \boldsymbol{\alpha})}dt = \mathbf{I}\end{aligned} \tag{16}$$

According to the Cauchy–Bunyakovsky–Schwarz inequality, the following can be derived

$$\begin{aligned}&\int_{-\infty}^{+\infty} (\hat{\boldsymbol{\alpha}} - \boldsymbol{\alpha}) \sqrt{f(t, \boldsymbol{\alpha})} \frac{\partial}{\partial \boldsymbol{\alpha}} (\ln f(t, \boldsymbol{\alpha})) \sqrt{f(t, \boldsymbol{\alpha})}dt \\ &\cdot \int_{-\infty}^{+\infty} (\hat{\boldsymbol{\alpha}} - \boldsymbol{\alpha}) \sqrt{f(t, \boldsymbol{\alpha})} \frac{\partial}{\partial \boldsymbol{\alpha}} (\ln f(t, \boldsymbol{\alpha})) \sqrt{f(t, \boldsymbol{\alpha})}dt \\ &\le var(\hat{\boldsymbol{\alpha}}) \cdot E[(\frac{\partial}{\partial \boldsymbol{\alpha}} (\ln f(t, \boldsymbol{\alpha})))^2]\end{aligned} \tag{17}$$

where $var(\hat{\boldsymbol{\alpha}}) = \int_{-\infty}^{+\infty} (\hat{\boldsymbol{\alpha}} - \boldsymbol{\alpha})^2 f(t, \boldsymbol{\alpha})dt$, $E[(\frac{\partial}{\partial \boldsymbol{\alpha}} (\ln f(t, \boldsymbol{\alpha})))^2] = \int_{-\infty}^{+\infty} (\frac{\partial}{\partial \boldsymbol{\alpha}} (\ln f(t, \boldsymbol{\alpha})))^2 f(t, \boldsymbol{\alpha})dt$. Hence, it has

$$var(\hat{\boldsymbol{\alpha}}) \cdot E[(\frac{\partial}{\partial \boldsymbol{\alpha}} (\ln f(t, \boldsymbol{\alpha})))^2] \ge \mathbf{I} \tag{18}$$

Now the CRLB $Fisher^{-1}(\boldsymbol{\alpha})$ of the motion analysis can be obtained:

$$var(\hat{\boldsymbol{\alpha}}) \geq Fisher^{-1}(\boldsymbol{\alpha}) \tag{19}$$

where $Fisher(\boldsymbol{\alpha}) = E[\frac{\partial}{\partial \boldsymbol{\alpha}}(\ln f(t, \boldsymbol{\alpha}))^2]$ is the Fisher information. The optimal is obtained when $\hat{\boldsymbol{\alpha}} - \boldsymbol{\alpha} = C\frac{\partial}{\partial \boldsymbol{\alpha}}(\ln f(t, \boldsymbol{\alpha}))$, where $C$ is a constant.

Now we can start to discretize the CRLB of motion analysis. Considering that the error vector $\mathbf{N}$ follows the Gaussian distributions, we have $\mathbf{N} = \mathbf{T} - \mathbf{H}\boldsymbol{\alpha} \sim N(0, \sigma^2 * \mathbf{I})$, and its joint probability distribution $F(\mathbf{T}, \boldsymbol{\alpha})$ is

$$
\begin{aligned}
F(\mathbf{T}, \boldsymbol{\alpha}) &= \Pi_{k=1}^n \frac{1}{\sqrt{2\pi}\sigma} exp\left(-\frac{(t_k - \theta_k \boldsymbol{\alpha})^T * (t_k - \theta_k \boldsymbol{\alpha})}{2\sigma^2}\right) \\
&= \left(\frac{1}{\sqrt{2\pi}\sigma}\right)^n exp\left(-\frac{(\mathbf{T} - \mathbf{H}\boldsymbol{\alpha})^T * (\mathbf{T} - \mathbf{H}\boldsymbol{\alpha})}{2\sigma^2}\right)
\end{aligned} \tag{20}
$$

Take the natural logarithm of Equation (20) and then compute the partial derivatives:

$$\frac{\partial}{\partial \boldsymbol{\alpha}} \ln F(\mathbf{T}, \boldsymbol{\alpha}) = \left(\frac{\mathbf{H}^T \mathbf{H}}{\sigma^2}\right)((\mathbf{H}^T \mathbf{H})^{-1} \mathbf{H}^T \mathbf{T} - \boldsymbol{\alpha}) \tag{21}$$

According to the optimal constraint of the Cauchy–Schwarz inequality, the best estimation of $\boldsymbol{\alpha}$ is obtained with

$$\hat{\boldsymbol{\alpha}} - \boldsymbol{\alpha} = C * \frac{\partial}{\partial \boldsymbol{\alpha}}(\ln F(\mathbf{T}, \boldsymbol{\alpha})) \tag{22}$$

Let $C$ be $\left(\frac{\mathbf{H}^T \mathbf{H}}{\sigma^2}\right)^{-1}$; then, the minimum variance unbiased estimate (MVUE) is obtained:

$$\hat{\boldsymbol{\alpha}} = (\mathbf{H}^T \mathbf{H})^{-1} \mathbf{H}^T \mathbf{T} \tag{23}$$

So we obtain

$$
\begin{pmatrix} \sum_{i=1}^n \theta_i^2 & \sum_{i=1}^n \theta_i \\ \sum_{i=1}^n \theta_i & n \end{pmatrix} \begin{pmatrix} k \\ b \end{pmatrix} = \begin{pmatrix} \sum_{i=1}^n \theta_i t_i \\ \sum_{i=1}^n t_i \end{pmatrix} \tag{24}
$$

The parameter $k$ therefore can be estimated via

$$k = \frac{\sum_{i=1}^n \theta_i \sum_{i=1}^n t_i - n \times \sum_{i=1}^n \theta_i t_i}{\sum_{i=1}^n \theta_i \sum_{i=1}^n \theta_i - n \sum_{i=1}^n \theta_i^2} \tag{25}$$

where $n$ is the count of snapshots including prior target information. Then, we need to make $\theta = \arctan(k)$, which is the tilt angle of the target trajectory.

*2.3. Template Generation*

This part illustrates the matching template generation using the Gabor filter. The Fourier transform is the most widely used transform in signal processing; however, it lacks local time and position information. Dennis Gabor introduced the renowned "window" Fourier transform, particularly short-time Fourier transform (STFT) [22], in the article "theory of communication" in 1946 in order to examine the frequency characteristics of the signal in the local range. The Gabor filter [23] is a short-time Fourier transform with a Gaussian window function that is extensively employed as a linear filter for edge enhancement in image processing. The two-dimensional form of the Gabor function is

$$
\begin{aligned}
&gabor_{2D}(x, y, x_0, y_0, \sigma_x, \sigma_y, u_x, u_y) \\
&= \rho e^{j\phi} s(x, y) l(x, y, \sigma_x, \sigma_y)
\end{aligned} \tag{26}
$$

with

$$l(x, y, \sigma_x, \sigma_y) = e^{-\pi((x-x_0)^2/\sigma_x^2 + (y-y_0)^2/\sigma_y^2)} \tag{27}$$

$$s(x, y) = e^{j(2\pi(u_x x + u_y y))} \tag{28}$$

where $(x_0, y_0)$ represents the center point of the Gaussian function, $\phi$ represents the initial phase, $(\sigma_x, \sigma_y)$ is the scale in two directions, $u_x$ and $u_y$ are frequency domain coordinates, and $\rho$ is the amplitude ratio of the Gaussian function. In a comprehensive way, the Gaussian function also has rotation (clockwise):

$$(x - x_0)_r = (x - x_0) \cos \varphi + (y - y_0) \sin \varphi \tag{29}$$

$$(y - y_0)_r = -(x - x_0) \sin \varphi + (y - y_0) \cos \varphi \tag{30}$$

where $\varphi$ is the rotation direction of the Gaussian function. The final Gabor function therefore can be rewritten as

$$gabor_{2D}(x, y, x_0, y_0, \varphi, \sigma_x, \sigma_y, u_x, u_y)$$
$$= \rho e^{-\pi((x-x_0)_r^2/\sigma_x^2 + (y-y_0)_r^2/\sigma_y^2)} e^{j(2\pi(u_x x + u_y y) + \phi)} \tag{31}$$

The real and imaginary components of a 2D Gabor filter are shown in Figure 3a,b, with the real component utilized to simulate the desired template. Furthermore, in this study, it is assumed that the template has the same scale $\sigma_x$ ($\sigma = \sigma_x = \sigma_y$) and frequency coefficient $u$ ($u = u_0 = v_0$) in both dimensions. The real component of the Gabor function is therefore expressed as

$$gabor_{real}(x, y, \lambda, \delta, \phi, \varphi, \gamma)$$
$$= exp(-\frac{x_r^2 + \gamma^2 y_r^2}{2\sigma^2}) \times \cos(\frac{2\pi x_r}{\lambda} + \phi) \tag{32}$$

with

$$x_r = x \cos \varphi + y \sin \varphi \tag{33}$$

$$y_r = -x \sin \varphi + y \cos \varphi \tag{34}$$

where $\gamma$ is the spatial aspect ratio of stripes, $\delta$ is the standard deviation of the Gaussian factor, $\lambda$ is the wavelength used to adjust the width of stripes. The transformations between parameters are such as $\frac{1}{2\delta^2} = \frac{1}{\sigma_x^2}$, $\frac{\gamma^2}{2\delta^2} = \frac{1}{\sigma_y^2}$.

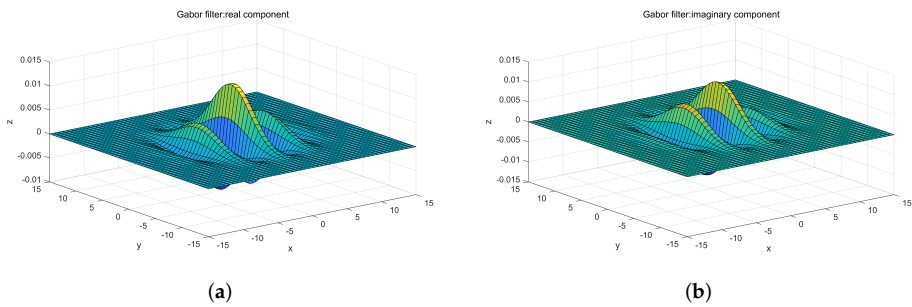

**Figure 3.** An example of Gabor filter. (**a**) Real component in the spatial domain. (**b**) Imaginary component in the spatial domain.

A 2D Gabor filter is the spatial domain product of a sinusoidal plane wave with a Gaussian kernel function. It has the ability to achieve excellent localization in both the spatial and frequency domains, which is similar to human biological visual features, allowing it to adequately characterize the local structural information of spatial frequency (scale),

spatial position, and direction selectivity. Assigning the time and frequency parameters $< \theta_r, t_r >$ to $< x_r, y_r >$, the final template generating model is described as

$$
\begin{aligned}
G(\theta, t, \lambda, \delta, \phi, \varphi, \gamma) &= gabor_{real}(\theta, t, \lambda, \delta, \phi, \varphi, \gamma) \\
&= exp(-\frac{\theta_r^2 + \gamma^2 t_r^2}{2\delta^2}) \times \cos(\frac{2\pi\theta_r}{\lambda} + \phi)
\end{aligned}
\tag{35}
$$

with

$$
\theta_r = \theta \cos \varphi + t \sin \varphi \tag{36}
$$

$$
t_r = -t \sin \varphi + \theta \cos \varphi \tag{37}
$$

Comparing (1) and (35), this study proposes that they possess coherence properties by setting the template parameters using the coherent matching of Gabor's zero point distribution and Rayleigh limit zero-point distribution. Therefore, Gabor functions are helpful for the decomposition of 2D pattern features by matching with BTR.

### 2.4. Two-Dimensional Template Matching

This section uses 2D template matching [2] to estimate the DOA of the interested targets. The impulse response and transfer function of the matched filter are assumed to be $\hbar(\theta, t)$ and $\mathbb{H}(u, v)$, respectively. As a result, the filter's output may be stated as

$$
y(\theta, t) = (I(\theta, t) + \bar{n}(\theta, t)) * \hbar(\theta, t) \tag{38}
$$

where $y(\theta, t)$ represents the output of the filter, $I(\theta, t)$ represents the baseband signal, and $\bar{n}(\theta, t)$ represents the additive noise in the image level. The power spectral density of $I(\theta, t)$ and image-level noise $\bar{n}(\theta, t)$ are denoted as

$$
\mathbb{F}(u, v) = \int_{-\infty}^{\infty} \int_{-\infty}^{\infty} I(\theta, t) e^{-j(u\theta + vt)} d\theta dt \tag{39}
$$

and

$$
\mathbb{N}(u, v) = \int_{-\infty}^{\infty} \int_{-\infty}^{\infty} \bar{n}(\theta, t) e^{-j(u\theta + vt)} d\theta dt \tag{40}
$$

where $u$ and $v$ correspond to the frequencies of the BTR within the different dimensions. With inverse Fourier transformation, the instantaneous output of the filter is rewritten as

$$
\begin{aligned}
y(\theta, t) = \int_{-\infty}^{\infty} \int_{-\infty}^{\infty} [\mathbb{H}(u, v)(\mathbb{F}(u, v) + \\
\mathbb{N}(u, v)) e^{j(u\theta + vt)}] du dv
\end{aligned}
\tag{41}
$$

Now we have the instantaneous output signal-to-noise ratio (SNR) of the filter as

$$
SNR(\theta, t) = \frac{[\int_{-\infty}^{\infty} \int_{-\infty}^{\infty} \mathbb{H}(u, v) \mathbb{F}(u, v) e^{j(u\theta + vt)} du dv]^2}{[\int_{-\infty}^{\infty} \int_{-\infty}^{\infty} \mathbb{H}(u, v) \mathbb{N}(u, v) e^{j(u\theta + v\tau)} du dv]^2} \tag{42}
$$

Suppose $\bar{n}(\theta, t)$ is the Gaussian white noise with the power density $N_o/2$, Equation (42) can be simplified to

$$
SNR(\theta, t) = \frac{2[\int_{-\infty}^{\infty} \int_{-\infty}^{\infty} \mathbb{H}(u, v) \mathbb{F}(u, v) e^{j(u\theta + vt)} du dv]^2}{N_o \int_{-\infty}^{\infty} \int_{-\infty}^{\infty} |\mathbb{H}(u, v)|^2 du dv} \tag{43}
$$

The optimal matched filter can be obtained by maximizing Equation (43). According to the Cauchy–Bunyakovsky–Schwarz inequality, we have

$$
[\int_{-\infty}^{\infty}\int_{-\infty}^{\infty}\mathbb{H}(u,v)\mathbb{F}(u,v)e^{j(u\theta+vt)}dudv]^2
$$
$$
\leq \int_{-\infty}^{\infty}\int_{-\infty}^{\infty}|\mathbb{H}(u,v)|^2dudv\int_{-\infty}^{\infty}\int_{-\infty}^{\infty}|\mathbb{F}(u,v)|^2dudv
$$

(44)

Hence, the optimal output SNR is

$$
SNR_{opt} = \frac{2}{N_o}\int_{-\infty}^{\infty}\int_{-\infty}^{\infty}|\mathbb{F}(u,v)|^2dudv
$$

(45)

The optimal SNR is achieved when

$$
\mathbb{H}(u,v) = C[\mathbb{F}(u,v)e^{j(u\theta+vt)}]^*
$$

(46)

## 3. Results

We evaluate this study firstly by analyzing Gabor filter templates in the simulations, and then using the 2D matching filtering characteristic. Two groups of tests are compared with and without ambient noise to validate the tracking capabilities with jamming targets. A series of quantitative studies are then used to assess the reliability of the target tracking systems under various SNR conditions. Finally, the proposed strategy is tested using real-world sea-trial data.

### 3.1. Evaluation of Gabor Filter Template

The produced Gabor templates $G$, whose x- and y-axes correspond to the bearing and time, are shown in Figure 4. The template library settings are based on the typical movement of actual targets at sea; therefore, it is assumed that the targets' maximum speed $v_{max}$ is 40 kn (about 74 km per hour), the system response time $T_{min}$ is 13 s, and the system's minimum effective distance $d_{min}$ is 2 km. The gray values correspond to the strength of the synthetic signals at $<\theta, t>$, and $G_i$ is the $i$-th template in the template library. The bearing range of $G_i$ is roughly determined as $\beta = (90 \times v_{max} \times T_{min})/(\pi \times d_{min}) \approx 3.83°$. To ensure easy calculation, $\beta$ is set to 4° in this experiment. According to the parameters mentioned above, we set the start bearing $\theta = (i-9) \times \Delta\theta_{step}$ and the end bearing $\theta' = (9-i) \times \Delta\theta_{step}$ for any $G_i$ ($i = 1, 2, 3, \cdots, 17$). This subsection then assesses the paper's Gabor template generating approach. The bearing resolution $\Delta\theta_{step}$ is 0.5 degrees, and the height of a single template is 13-snapshot intervals comparable to $T_{min} = 13$ s, and the spatial aspect ratio $\gamma$ of the stripes is 0.05, indicating that the Gabor function should extend along the time axis. As illustrated in Equation (35), the Gabor function is a Gaussian function modulated by a sinusoidal function ($\cos(\frac{2\pi\theta_r}{\lambda} + \phi)$), and the initial phase is zero; therefore, the first null point $\theta_{firstNull}$ is determined by $\frac{2\pi\theta_{firstNull}}{\lambda}$. Let $pi/2 = \frac{2\pi\theta_{firstnull}}{\lambda}$, then $\theta_{firstnull} = lamda/4$; therefore, if we want the template width to be 3, the stripe width $\lambda$ will be $3 \times 4 = 12$. The energy optimum of a Gaussian function is in the $[-3\sigma, 3\sigma]$ range; thus, if $3\sigma = \lambda/4$, we obtain $\sigma = \lambda/12$, and the standard deviation $\delta$ of the Gaussian component is $\lambda/12 = 12/12 = 1$, as indicated in Equation (35).

Under the assumption of a linear model, the two-dimensional time-bearing template-library set above covers all target-bearing modes; that is, when the prior TMA parameters are specified, Gabor filtering can generate the corresponding matching BTR trajectory within the set time.

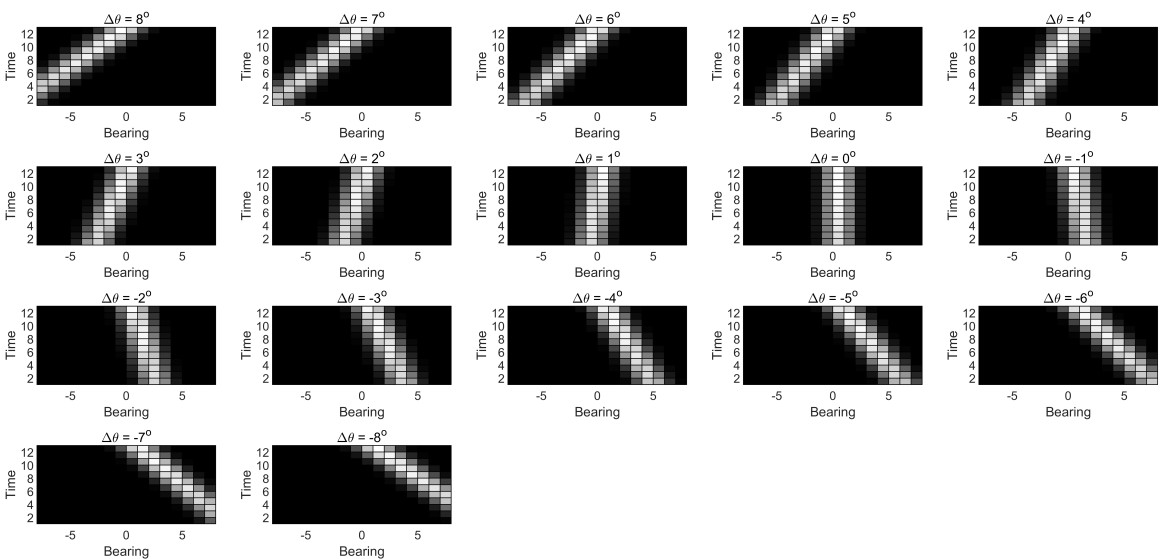

**Figure 4.** Gabor template library.

### 3.2. Comparison Experiments with Jamming Targets

Figure 5 shows that the noiseless BTR diagram's bearing space starts from $-180°$ and ends at $+180°$, and the resolution step is $0.5°$. Brighter trajectories indicate more potent energy, and it is assumed that five acoustic targets were collected. The first and fourth targets have lower normalized power levels than others. In order to emphasise the performance of the proposed algorithm, the two weak targets are chosen as the interested targets with ISR = $-15$ db. Then we conduct three implementations: traditional maximum energy tracking (ME), linear template matching (LTM) as reported in [21], and the proposed technique.

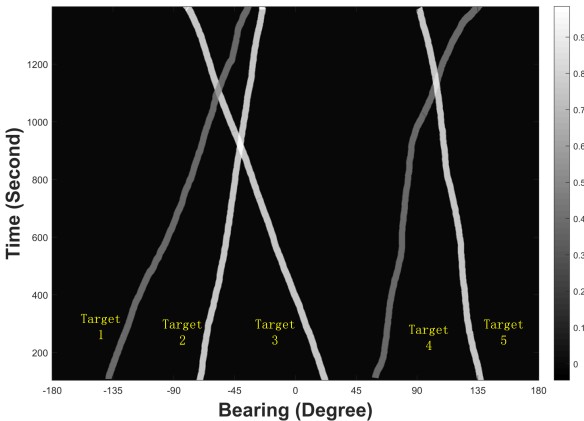

**Figure 5.** Original Bearing-Time Record.

The tracking results are depicted in Figure 6. For target motion analysis, 90 snapshots (equal to a 90 s data set) are employed. To begin, we do not apply Gaussian white noise to the simulated BTR. When using ME, as shown in Figure 6a,d, tracking bias and tracking failure are induced when the interested and interfering target trajectories intersect, and the proposed method or LTM can suppress this interference to a certain extent. However, Figure 6b,e show that the trajectory is broken, whereas the tracking results using the proposed method are smooth and continuous near the jamming part, as shown in Figure 6c,f. As a result, it can be inferred that the approach described in this study is capable of jamming-target suppression when compared to previous methods.

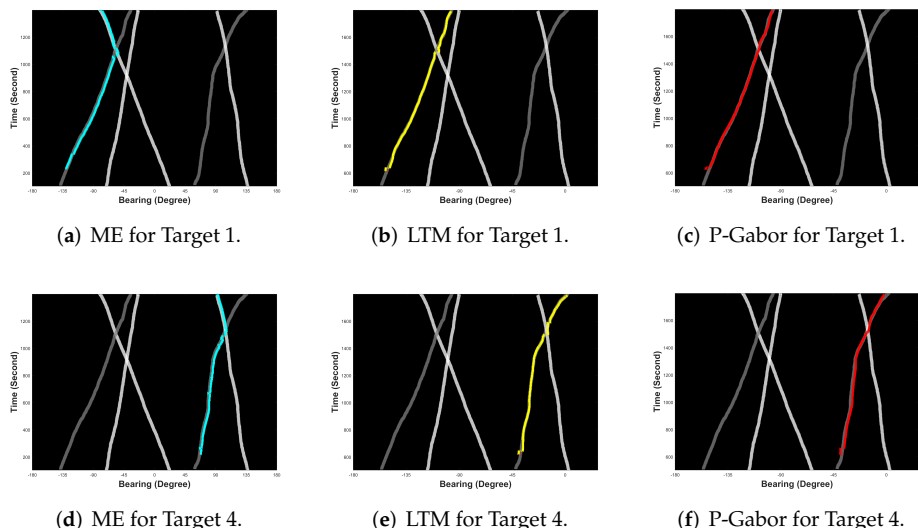

(**a**) ME for Target 1.      (**b**) LTM for Target 1.      (**c**) P-Gabor for Target 1.

(**d**) ME for Target 4.      (**e**) LTM for Target 4.      (**f**) P-Gabor for Target 4.

**Figure 6.** Experiment results with jamming targets.

We extract a single snapshot at the jamming point on the BTR to display and explain the processing impact of the method more clearly, and the bearing spectrum before and after processing is presented in Figure 7. The noisy target is clearly muted, while the interesting one is amplified to a greater energy level. As a result, when the search window searches for the desired target on the BTR, the tracking success rate is considerably boosted.

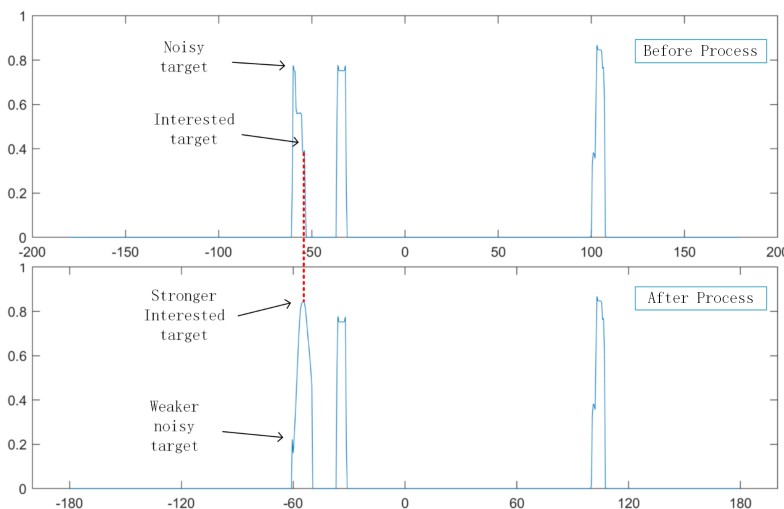

**Figure 7.** A single snap of BTR.

### 3.3. Reliability Evaluations

This subsection's experiment assesses the reliability of the proposed method in different SNR and SIR situations. The SNR value is defined as

$$SNR = 10 \times \log(P_s / P_n) \tag{47}$$

where $P_s$ and $P_n$ are the image level powers of signals and ambient noise. Gaussian white noise is used as the ambient noise. Figure 8 shows the evaluation results, and the LTM is evaluated in the first column of the sub-figures at SNR = $-21.65$ dB ($\sigma = 0.85$); we can see that the targets that are being monitored have failed. However, as SNR is further dropped, as demonstrated in the second- and third-column sub-figures, Targets 1 and 4 are lost until $-25.11$ dB ($\sigma = 1.25$).

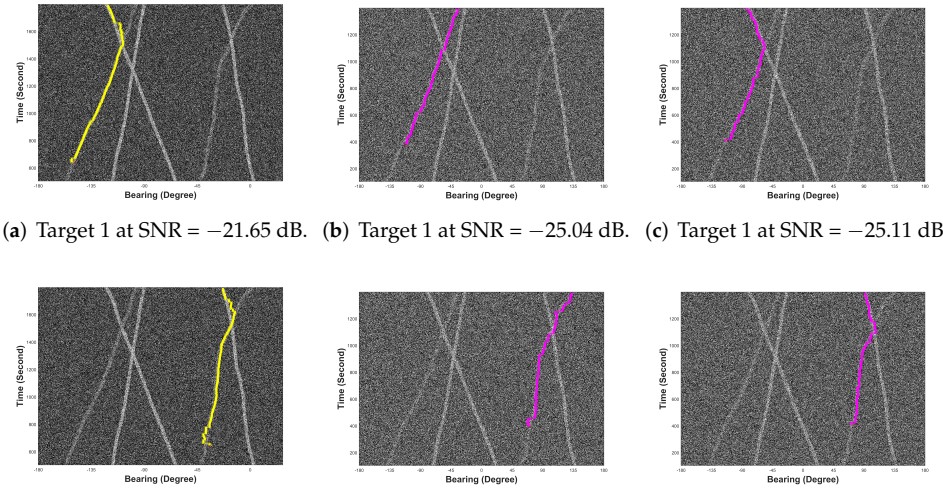

(**a**) Target 1 at SNR = −21.65 dB. (**b**) Target 1 at SNR = −25.04 dB. (**c**) Target 1 at SNR = −25.11 dB.

(**d**) Target 4 at SNR = −21.65 dB. (**e**) Target 4 at SNR = −25.04 dB. (**f**) Target 4 at SNR = −25.11 dB.

**Figure 8.** Tracking results of the LTM (yellow line) and Proposed Method (purple line) over different SNRs.

The SIR boundary of the algorithm is evaluated by using the data shown in Figure 9a in which there are three target trajectories. From the starting point, the second trajectory from the left has the weakest energy, SIR= −25 dB, background SNR = 0 dB, and there is a clear intersection with the strong target. As shown in Figure 9b, we successfully track it using the proposed method, i.e., reaching the performance limit SIR = −23 dB.

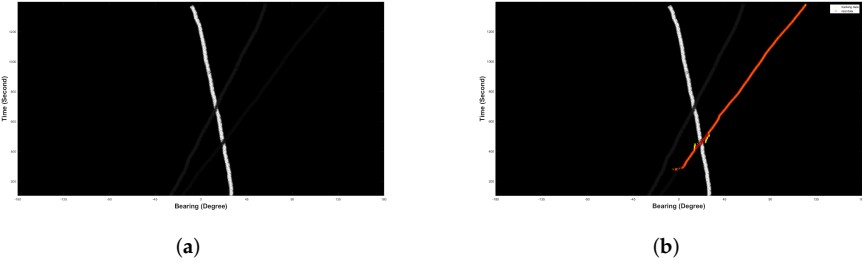

(**a**) (**b**)

**Figure 9.** Tracking results at SIR = −23 dB. (**a**) Jamming target at SIR = −23 dB. (**b**) Tracking result with proposed method.

The results of this subsection's experiments demonstrate that the suggested approach can track an interested target stably despite jamming-target interference and has the ability to suppress noise.

*3.4. Accuracy Performance Evaluations*

This part compares the accuracy performance of the proposed Gabor tracking approach to that of PCA-only tracking. The data base is the BTR picture shown in Figure 10, in which the genuine trajectories are indicated with blue dots and the tracking results with red dots. The performance metric is the mean absolute bearing deviations of three separate runs.

The tracking data are compared in Figure 11. The PCA-only curve carrying error may be approximated to a horizontal line, illustrating that PCA-only migrates the target tracing. The mean absolute deviations of the proposed implementation diminish linearly as SNRs increase. It should be emphasized, however, that these tracking results are predicated on the assumption that the target is effectively and stably tracked.

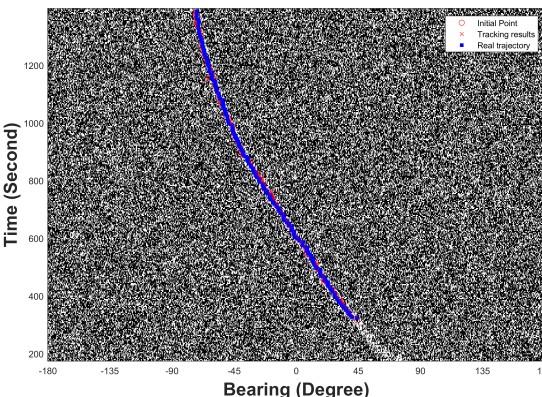

**Figure 10.** Example of Evaluation Experiments.

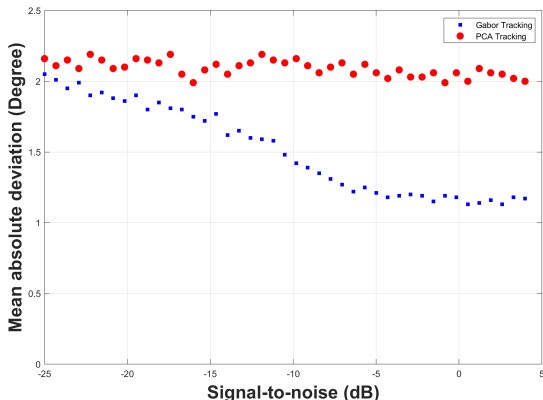

**Figure 11.** Evaluation of Accuracy Performance.

### 3.5. Tracking Results with Sea-Trial Data

Following those simulation tests, we employ real sea-trial data to evaluate and validate the proposed approach. The sea trial is conducted by the State Key Laboratory of Acoustics, Institute of Acoustics, Chinese Academy of Sciences on the South China Sea. The test ship moves in variant distances and directions, and its signal receiver is a horizontal array composed of 128 array elements. The sound velocity profile is provided in Figure 12a to characterize the circumstances surrounding the sea region of experiment.

Figure 12d shows that five acoustic targets were collected and labeled with a number. The first and second targets appear to have higher normalized power levels than the other three targets.

The proposed approach is next tested against the LTM in a series of comparison trials. To begin, consider Targets 1 and 2 to be the interested targets, while the remainder are jamming targets. The noise power of the trajectories to track is larger in this case, making the work easier. As predicted, Figure 12c,f indicate that the proposed approach can successfully track the interested targets. Figure 12b,e reveal that the LTM can follow the targets as well, although its accuracy appears to be worse than the former. We can observe that the LTM's tracking constantly deviates. The SIRs of Targets 4 and 5 are −19.1 dB and −17.8 dB, respectively, with noise powers lower than the jamming; Figure 13d,e show that the proposed approach can still track the target successfully, but Target 3, whose SIR is −22.3 dB, fails, as shown in Figure 13a–c. Figure 13f, on the other hand, displays an unexpected tracking result. The tracking trajectory deviates under interference from the first jamming target. Figure 14 zooms in Figure 13f for a further analysis, and it shows that the interested target is completely covered by the two jamming targets, whose trajectories are extremely close around this part. From the perspective of the image, the interested target cannot be detected in the next dozens of snapshots, making tracking impossible using

the template matching approach. On the other hand, the power levels of this interested target are quite weak compared with these two jamming ones, which means the SNR is too low to track.

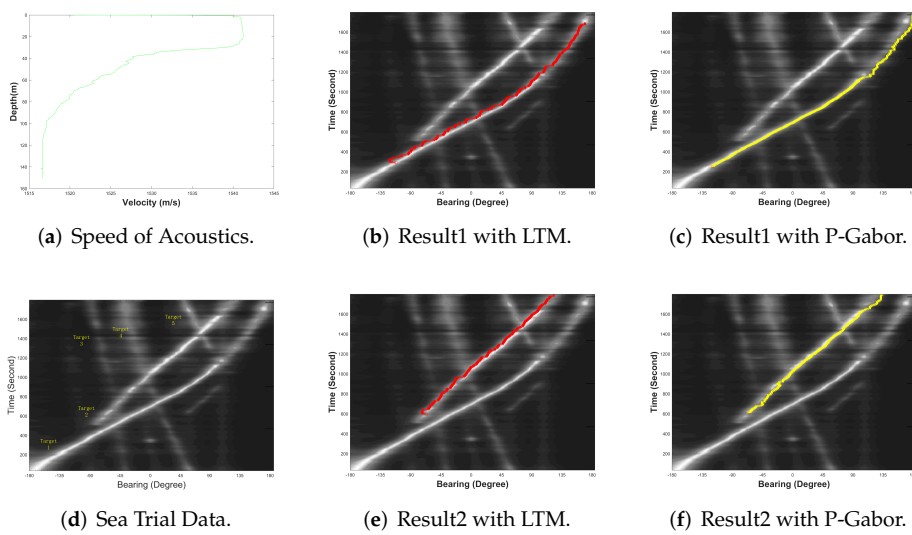

(**a**) Speed of Acoustics.  (**b**) Result1 with LTM.  (**c**) Result1 with P-Gabor.

(**d**) Sea Trial Data.  (**e**) Result2 with LTM.  (**f**) Result2 with P-Gabor.

**Figure 12.** Experiment results with weak jamming targets.

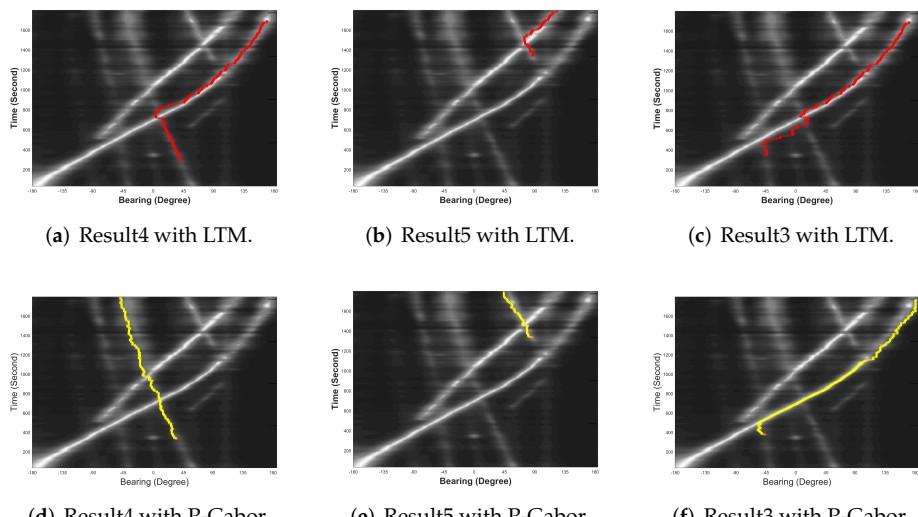

(**a**) Result4 with LTM.  (**b**) Result5 with LTM.  (**c**) Result3 with LTM.

(**d**) Result4 with P-Gabor.  (**e**) Result5 with P-Gabor.  (**f**) Result3 with P-Gabor.

**Figure 13.** Experiment results with strong jamming targets.

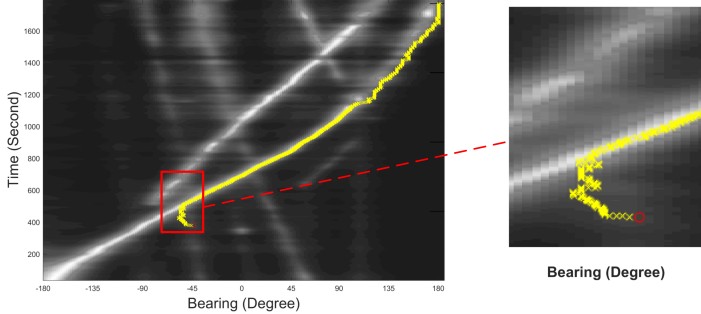

**Figure 14.** Observation Window of jamming area.

## 4. Conclusions

This paper proposes a weak-target-tracking algorithm under jamming circumstances based on image processing, which we call 'P-Gabor'. The experiment results show that the presented approach can successfully realize automatic target tracking with jamming targets and is proved to have certain capabilities of noise suppression. First, we denoise the input BTR using the PCA approach, which improves tracking performance under low SNR conditions. Then, we performed TMA on the BTR diagram by using linear fitting, which can offer a motion prediction in the following snapshots. The proposed technique then constructs the 2D matching template using the Gabor function based on the estimated parameters above. A practical target-trajectory-template-generating approach is proposed, allowing the template set to be customized for various sensor array configurations. The templates are then compared with the target trajectories to obtain the bearing of the interested target. Next, we do a series of simulations to testify that the algorithm can track weak target with strong jamming targets successfully under low SIR to −23 dB. Finally, a more difficult scenario with low SNR is performed in order to assess the dependability of the proposed technique. Furthermore, to validate the proposed approach's tracking capabilities under low SNR, we compare it to the LTM methodology in [21], which performed much better under low SNR conditions even with −25 dB. Moreover, as demonstrated in Figure 11, the proposed technique outperforms the present PCA tracking method in terms of accuracy performance. However, we can observe from the assessed tests in sea trials that when the interested target is completely covered by the two jamming targets, it is impossible to track effectively using the template-matching approach.

In future work, some improvements will be made regarding the accuracy of azimuth tracking in various ways.

**Author Contributions:** F.Y. (Fan Yin): Methodology, software, methodology, writting, investigation, writing—original draft preparation; C.L.: Investigation, resources, data curation, project administration and funding acquisition; H.W.: Investigation, resources, writing—review and editing, supervision, project administration and funding acquisition; F.Y. (Fan Yang): Investigation, conceptualiuzation, writing—review and editing. All authors have read and agreed to the published version of the manuscript.

**Funding:** This work was supported by the China Scholarship Council, Chinese Academy of Sciences, and National Natural Science Foundation of China (ID: 62171440).

**Acknowledgments:** The authors would like to thank the editors and the reviewers for their comments on the manuscript of this article.

**Conflicts of Interest:** The authors declare that they have no competing interests.

## Abbreviations

The following abbreviations are used in this manuscript:

| | |
|---|---|
| DOA | Direction of arrival |
| BTR | Bearing time records |
| CAC | China Automation Congress |
| GPU | Graphics processing unit |
| CBF | Conventional Beamforming |
| SNR | Signal-to-noise ratio |
| PCA | Principal component analysis |
| SVD | Singular value decomposition |
| CRLB | Cramer-Rao Lower Band |
| TMA | Target motion analyzing |
| MVUE | Minimum-variance unbiased estimator |
| STFT | Short-time Fourier transform |
| ME | maximum energy tracking |
| LTM | Linear template matching |

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
