# Peer review of "Automatic Tracking of Weak Acoustic Targets within Jamming Environment by Using Image Processing Methods"

_applsci, doi:10.3390/app12136698_

Round 1

Reviewer 1 Report

The manuscript presents a tracking framework based on visual pattern recognition algorithms, which includes target motion analysis.

The background of this research is well explained. The problem formulation is considered to be fine. The proposed method also judged to be adequate. However, th eexperimental results are qualitative, rather than quantitative.

The authors should disclose quantitative results, which are generally apprecable.

Reviewer 2 Report

In the article " Automatic tracking of underwater acoustic weak targets disturbed by strong jamming targets under the framework of image processing", the authors presents a tracking framework based on visual pattern recognition algorithms, which includes target motion analysis, matched filtering, and principal component analysis-based denoise.

In the introduction, the authors indicated that one of the most important research topics in matrix signal processing, commonly known as spatial spectrum estimation, is the direction of arrival (DOA). When multiple DOA spectrograms are collected, a bearing time recording (BTR) is plotted on the Timeline. However, BTR tracking is often blocked by disruptive targets. Traditional BTR tracking methods rely on maximum energy tracking, which means that if the noise level of the target concerned is less than jamming, they inevitably fail. As a result, it is imperative to develop mitigation strategies against powerful disruptive targets. The authors indicated as the main goals of the article:

a) insufficient research in the field of BTR post-processing. This article establishes a general BTR post-processing architecture that can be used to improve the resolution and traceability of BTR.

b) addressing the tracking failure caused by BTR image level noise by introducing a real-time PCA processing method and proposing a method to improve BTR trajectory to improve the stability of traditional target tracking methods.

c) solving the insufficient resolution problem caused by the Rayleigh BTR limit by the BTR resolution improvement algorithm by combining the target motion analysis with the Gabor filtering method.

d) obtaining a higher fit gain by setting the template parameters using a coherent fit of Gabor zero-point distribution and Rayleigh limit zero point distribution.

In chapter 2, the authors describe the proposed approach to the acoustic tracking of the target's trajectory. Authors using principal component analysis (PCA) first denoise the original BTR to improve tracking performance. The target traffic analysis parameters are then derived using the prior linear fit information. And then they use the Gabor function to construct a match template based on the estimated parameters. Finally, the target bearing is estimated by matching the templates to the target trajectory.

In Chapter 3, the authors of the tracking experiments first analyzed using Gabor filter templates and then assessed using the 2-D fit filtering characteristics. They compared two sets of tests with and without ambient noise to verify traceability with disruptive targets. They then applied a series of quantitative studies to evaluate the reliability of target tracking systems under various SNR conditions. Finally, the proposed strategy was tested with real sea trial data.

Chapter 4 presents the results of the experiment. Based on image processing, the article proposes an approach to underwater acoustic tracking of bearing on targets with jamming targets. The results of the experiment show that the presented approach can successfully implement automatic target tracking with jamming targets and it has been proven that it has some noise suppression abilities. First, the input BTR has been denuded by the PCA method, which improves the tracking performance under low SNR conditions. Then, using a linear fit, an analysis of the target's motion was performed on the tracking time record, which enables the prediction of motion on subsequent snapshots. The proposed technique then constructs a two-dimensional fitting pattern using the Gabor function based on the parameters estimated above. A practical approach to generating target trajectory templates was proposed, allowing the template set to be adapted to different sensor matrix configurations. The templates are then compared to the target's trajectories to get a bearing on the target of interest. Finally, a more difficult scenario with a low SNR is performed to assess the reliability of the proposed technique. In addition, to verify the traceability of the proposed approach under low SNR conditions, it was compared to the LTM methodology which performed much better under low SNR conditions.

Comments:

The article covers the field and is correctly structured. The article is scientifically substantiated and the experiment was carried out correctly. The conducted experiment showed that the proposed technique is more efficient and accurate than the current method of PCA tracking. However, on the basis of the assessed sea trials, it can be observed that when the target is completely covered with a cork from two targets, it is impossible to effectively track using the pattern matching approach, which the authors will solve in further research, and then it will be possible to evaluate the method in its entirety.

At this time, it should be noted that most bibliographies are more than 5 years old. Extend the literature review to include items from the last five years.

Reviewer 3 Report

The research presents an underwater acoustic target bearing tracking approach with jamming targets based on image processing. Experimental results show that the presented approach can be successfully realize automatic target tracking with jamming targets and is proved to have certain capability of noise suppression. The article is well written, however there are some major issues which need to be addressed to improve the quality of publication.

1.       The Title of the article is too long.

2.       Abstract is too short. The distribution of the abstract should be as: (i) Research problem, (ii) Research methodology, (iii) Results and Findings, (iv) Conclusion / recommendations.

3.       Please avoid words “most” significant etc., in the introduction (line 10).

6.       Some background information should be provided at the start of the introduction section. 

4.       In the text Fig., is used however Figure is used in the actual figures, please use format provided by the journal.

6.       The studies from line 31 to 66 are mixed, should be explained separately with their limitations.   

5.       The innovations claimed in the introduction section should be properly highlighted in the following sections.

6.       It is suggested that the “Results and Discussion” should be merged and “Conclusion” should be a separate section.

Round 2

Reviewer 1 Report

Done.

Author Response

Done. Thanks for your suggestions!

Reviewer 3 Report

Many observations have been addressed in the revised version, however following corrections are also required to improve the quality of the publication.

Line 110: Section 2 describes the proposed tracking method ....

Line 113: Algorithm is used in the heading (please use the same heading title)

Line 114: This chapter describes..... (this is not chapter)

Line 114: seen in Figure 2. (Figure 2 has so many parts from a to g), should be explained in the text.

Line 139: slope et al. (please provide reference here)

The result section is starting from figure which is not standard practice. The figure(s) should be explained first.

Please check line 188, 189, the sentence should be continue in above line.

In conclusion one reference is used and one figure is referred, which is not standard practice.

Please check line no. 353, some ..... are in the middle of the sentence.

Line 110: Section 2 describes the proposed tracking method ....

Line 113: Algorithm is used in the heading (please use the same heading title)

Line 114: This chapter describes..... (this is not chapter)

Line 114: seen in Figure 2. (Figure 2 has so many parts from a to g), should be explained in the text.

Line 139: slope et al. (please provide reference here)

The result section is starting from figure which is not standard practice. The figure(s) should be explained first.

Please check line 188, 189, the sentence should be continue in above line.

In conclusion one reference is used and one figure is referred, which is not standard practice.

Please check line no. 353, some ..... are in the middle of the sentence.
